# Investigating Multilingual Coreference Resolution by Universal Annotations

**Haixia Chai** and **Michael Strube**
Heidelberg Institute for Theoretical Studies gGmbH
Schloss-Wolfsbrunnenweg 35
69118 Heidelberg, Germany
{haixia.chai, michael.strube}@h-its.org

## Abstract

Multilingual coreference resolution (MCR) has been a long-standing and challenging task. With the newly proposed multilingual coreference dataset, CorefUD (Nedoluzhko et al., 2022), we conduct an investigation into the task by using its harmonized universal morphosyntactic and coreference annotations. First, we study coreference by examining the ground truth data at different linguistic levels, namely mention, entity and document levels, and across different genres, to gain insights into the characteristics of coreference across multiple languages. Second, we perform an error analysis of the most challenging cases that the SotA system fails to resolve in the CRAC 2022 shared task using the universal annotations. Last, based on this analysis, we extract features from universal morphosyntactic annotations and integrate these features into a baseline system to assess their potential benefits for the MCR task. Our results show that our best configuration of features improves the baseline by 0.9% F1 score.[1]

## 1 Introduction

Coreference resolution is the task to identify expressions in a given text that refer to the same entity. While considerable progress has been made in coreference resolution for English (Lee et al., 2017, 2018; Joshi et al., 2019, 2020; Kirstain et al., 2021; Grenander et al., 2022), extending this task to multiple languages presents significant challenges due to the linguistic diversity and complexity of different languages. The multilingual coreference resolution (MCR) task (Recasens et al., 2010; Pradhan et al., 2012) focuses on developing a general and robust system that can effectively handle multiple languages and a wide range of coreference phenomena (e.g., pronoun-drop).

Recently, Nedoluzhko et al. (2022) propose a new set of multilingual coreference datasets, Coref-UD, built upon the framework of Universal Dependencies[2] (de Marneffe et al., 2021), allowing coreference researchers to conduct cross-linguistic studies across 17 datasets for 12 languages. The datasets serve as resource for the CRAC 2022 shared task on multilingual coreference resolution (Žabokrtský and Ogrodniczuk, 2022). Given the harmonized universal morphosyntactic and coreference annotations, we raise the question whether there are any universal features that are common to all languages and to what extent they can contribute to the development of an MCR system.

In this work, we conduct an in-depth investigation into the MCR task by using universal annotations in CorefUD. First, we analyze ground truth data from different linguistic levels, including mention, entity and document levels, and across different genres, to gain an understanding of coreference across various languages. Second, we conduct an error analysis of the most challenging cases that MCR systems fail to resolve. Last, based on this analysis, we integrate several features extracted from universal morphosyntactic annotations into a baseline system to examine their potential benefits for the MCR task. To the best of our knowledge, our method represents the first attempt to leverage universal annotations for MCR.

Our findings reveal: (i) There are indeed commonalities across languages. For example, we observe a common pattern where the closest antecedent of an overt pronoun mainly corresponds to the subject or object position. These commonalities are valuable for potential future research, such as linguistic investigations aimed at further comprehending the linguistic phenomenon of coreference. However, it is important to note that explor-

---

[1] Our code and model are publicly available at https://github.com/HaixiaChai/multi-coref.

[2] One of the benefits of Universal Dependencies is that it provides cross-linguistic guidelines for morphosyntactic annotation in a consistent and language-independent manner.

ing universal features is a challenging task due to the inherent variability among languages, e.g., the expression of definiteness. (ii) A common issue encountered in all languages by MCR systems is the difficulty of correctly detecting nominal nouns within some two-mention entities. (iii) Our experimental results show that our best configuration of features improves the baseline by 0.9% F1 score.

## 2 Related Work

**Analysis in Multiple Languages.** Coreference is a complex linguistic phenomenon that requires linguistic expertise, even more so when studying it in a multilingual context. Oftentimes, researchers primarily focus on investigating coreference within a single target language in which they possess expertise, enabling them to gain valuable insights specific to that language (Ogrodniczuk and Niton, 2017; Urbizu et al., 2019; Sundar Ram and Lalitha Devi, 2020). However, a few studies have been conducted on coreference across multiple languages by using multilingual coreference datasets (Recasens et al., 2010; Pradhan et al., 2012; Nedoluzhko et al., 2022). These studies include statistical analysis of the datasets (Nedoluzhko et al., 2022), as well as efforts to improve the performance and generalizability of MCR systems from a technical standpoint (Kobdani and Schütze, 2010; Björkelund and Kuhn, 2014; Straka and Straková, 2022). It is apparent that analyzing coreference across multiple languages is a challenging task due to the expertise required of each language. However, CorefUD helps such analyses by providing universal annotations. Our work is the first attempt to analyze cross-linguistic patterns and gain a broader understanding of coreference across different languages and language families in a comprehensive and comparative manner.

In the field of MCR, there has been notable attention directed towards the research of two types of languages. One prominent area of investigation is around pro-drop languages, such as Chinese (Kong and Ng, 2013; Song et al., 2020; Chen et al., 2021; Zhang et al., 2022), Italian (Iida and Poesio, 2011) and Arabic (Aloraini and Poesio, 2020, 2021). Another research direction involves the study of morphologically rich languages, such as German and Arabic (Roesiger and Kuhn, 2016; Aloraini and Poesio, 2021). In contrast to the aforementioned work, which primarily focuses on enhancing the model's capabilities through technical analysis of

specific linguistic phenomena, our research delves into gold annotations to explore multilingual coreference including phenomena like zero pronouns from a linguistic perspective, uncovering valuable insights to foster further research.

**MCR Systems.** In the past decade, numerous MCR approaches have been proposed, including rule-based approaches, various training methodologies such as cross-lingual and joint training, and methods that leverage linguistic information: (i) Rule-based. It requires a complete redefinition of a set of rules to transform a monolingual coreference resolution system into a multilingual one, for example when using Stanford's multi-pass sieve CR system (Lee et al., 2011). The adaptation process is time-consuming and requires a language expert to develop the rules. (ii) Translation-based projection. This is a technique that involves the automatic transfer of coreference annotations from a resource-rich language to a low-resource language using parallel corpora (Rahman and Ng, 2012; Chen and Ng, 2014; Martins, 2015; Novák et al., 2017; Lapshinova-Koltunski et al., 2019). The primary challenge of this approach is the occurrence of a large number of projected errors, such as a nominal phrase in English is translated as a pronoun in German. (iii) Latent structure learning. Fernandes et al. (2012) and Björkelund and Kuhn (2014) use a latent structure perceptron algorithm to predict document trees that are not provided in the training data. These document trees represent clusters using directed trees over mentions. This approach has achieved the best results in the CoNLL-2012 shared task for English, Chinese and Arabic at that time. (iv) Joint training. This is a technique that finetunes multilingual word embeddings on the concatenation of the training data in multiple languages. It allows the model to learn shared representations and help in cases where the target language has limited training data. (Straka and Straková, 2022; Pražák et al., 2021; Pražák and Konopik, 2022) (v) Methods with linguistic information. Several studies have incorporated syntactic and semantic information into their models (Zhou et al., 2011; Jiang and Cohn, 2021; Tan et al., 2021). These works either focus on coreference resolution within a single language or employ machine learning approaches to address the MCR task. Different from the above, our work incorporates universal morphosyntactic information into an end-to-end joint training method across multiple languages.

# 3 Linguistic Analyses on CorefUD 1.1

CorefUD 1.1[3] is the latest version of CorefUD (Nedoluzhko et al., 2022) for the CRAC 2023 shared task on multilingual coreference resolution, including 17 datasets for 12 languages.[4] In the following subsections, we conduct a linguistic study on it by using the ground truth from the training datasets, examining coreference phenomena from different linguistic levels, namely mention, entity and document perspectives, and across different genres, in multiple languages.

## 3.1 Mention

A mention is the smallest unit within a coreference relation, comprising one or more words (maybe even less than a word in some cases).

**Position of Head.** The head of a mention typically represents the entity being referred to. The remaining words in the mention either provide additional information that precedes the head word (pre-modification, e.g., *a highly radioactive element*) or further specify the meaning of the head after it (post-modification, e.g., *a car with leather seats.*). Note that the modifying words can be dominant in the mention in some cases, e.g., *the first floor*, making resolution of those mentions harder sometimes.

| ca | cs | en | hu | pl | es |
|------|------|------|--------|------|--------|
| 13% | 22% | 27% | **51%** | 14% | 14% |
| lt | fr | de | ru | no | tr |
| **52%** | 28% | 32% | 24% | 18% | **52%** |

Table 1: Percentage of pre-modified mentions in the respective languages.

Table 1 shows that Hungarian, Lithuanian and Turkish all have a high percentage of pre-modified mentions. They are from the Uralic, Baltic and Turkic language families that are considerably different from the other languages.

**Mention Types.** To gain insight into how mentions represent and refer to entities, we categorize five types of mentions by the universal part-of-speech (UPOS) tags of the head words in gold mentions, namely nominal noun, proper noun, overt pronoun, zero pronoun and others.

Unsurprisingly, in Figure 1, we observe that nominal noun and proper noun are the two

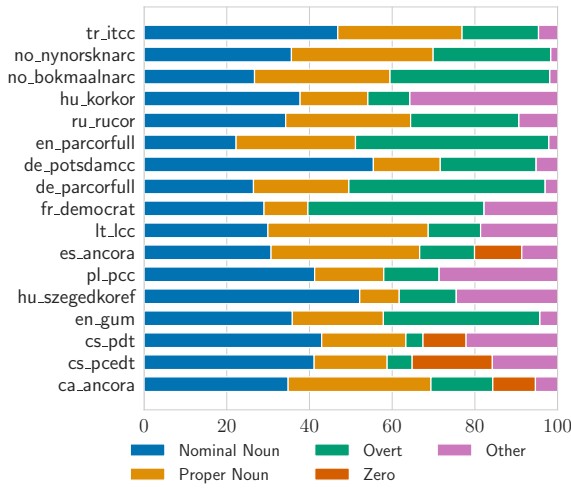

Figure 1: Percentage of mention types in datasets.

main categories of mentions in most of datasets. EN_PARCORFULL (Lapshinova-Koltunski et al., 2018), DE_PARCORFULL (Lapshinova-Koltunski et al., 2018) and FR_DEMOCRAT (Landragin, 2021) are the datasets having most overt pronouns, around 46% of mentions. In contrast, resolving zero pronouns is more crucial in the Czech datasets (Nedoluzhko et al., 2016; Hajič et al., 2020) where the number of zero pronouns is higher than that of overt pronouns.

**Universal Dependency Categories.** By using universal dependency (UD) relations between words in a sentence, we can understand the hierarchical structure of the sentence and identify the potential antecedents of referring expressions. We classify UD relations of heads of gold mentions into 12 categories according to the UD taxonomy[5], as illustrated in Table 2.

**Anaphor-Antecedent Relation.** Given mention types and UD categories presented above, we have a particular interest in analyzing the UD category of the closest antecedent to an anaphor based on its mention types (e.g., core arguments_subject - overt pronoun). We consider all mentions in an entity as observed anaphors, but exclude the first mention.

The results in Figure 2 present the UD relations that are most frequently associated with nominal noun and overt pronoun. We found that non-core dependents (e.g., oblique nominal), nominal dependents (e.g., numeric mod-

---

[3] See Appendix A for the statistics of CorefUD 1.1.
[4] https://ufal.mff.cuni.cz/corefud/crac23

[5] https://universaldependencies.org/u/dep/index.html

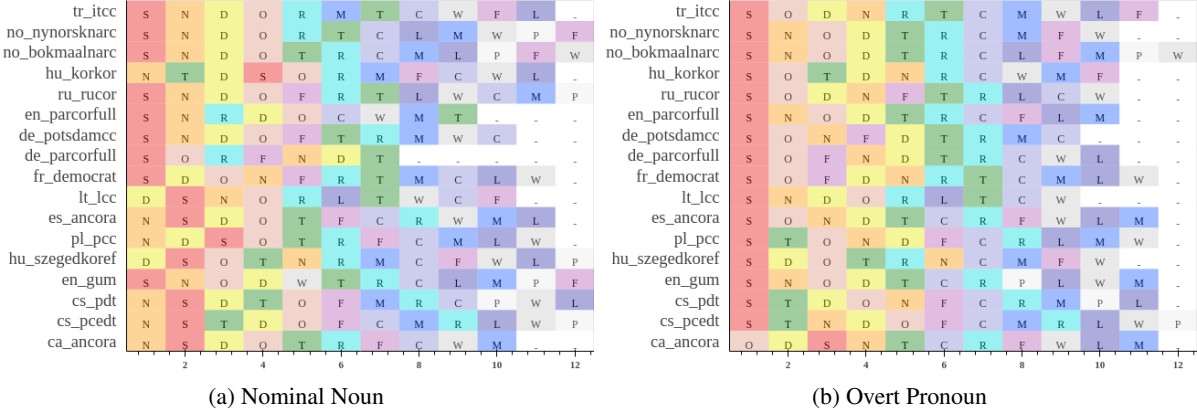

Figure 2: Ranking of relations between UD categories of antecedents and mention types of anaphors in each dataset. For instance, figure (a) shows, in each row, UD categories (initials) are ordered according to their frequency of association with nominal noun on a dataset. See Table 2 for the full name of UD categories.

| UD CATEGORIES | UD RELATIONS |
|---|---|
| core arguments_ subject (S) | nsubj |
| core arguments_ object (O) | obj, iobj |
| non-core dependents_ nominals (D) | obl, vocative, expl, dislocated |
| nominal dependents_ nominals (N) | nmod, appos, nummod |
| clauses (C) | csubj, ccomp, xcomp, advcl, acl |
| modifier words (M) | advmod, discourse, amod |
| function words (F) | aux, cop mark det, clf, case |
| coordination (R) | conj, cc |
| MWE (W) | fixed, flat, compound |
| loose (L) | list, parataxis |
| special (P) | orphan, goeswith, reparandum |
| other (T) | punct, root, dep |

Table 2: Universal dependency categories.

ifier, nominal modifier and appositional modifier), `core arguments_subject` and `core arguments_object` are the primary UD relations of antecedents for nominal noun, e.g., *Sam, my brother, John 's cousin, arrived.* In contrast, the closest antecedents of overt pronoun mainly correspond to subjects or objects within core arguments.[6] It is important that these findings are applicable across all languages, emphasizing their universal relevance in the context of the multilingual coreference resolution task.

## 3.2 Entity

In a text, an entity can have multiple mentions all referring to the same identifiable object, such as a

person or concept. Each gold entity in all datasets of CorefUD 1.1 has 3 to 4 mentions on average without considering singletons.

**First Mention.** The first mention within a mention chain serves to introduce the entity into a context. Thus, this mention could be seen as the most informative expression in the entity. In CA_ANCORA (Recasens and Martí, 2010), for example, 97% of first mentions belong to mention types of nominal noun or proper noun, which convey a richer semantic meaning than pronouns. Furthermore, we observe a consistent trend across all languages that the ratio of entities with the first mention being the longest mention in the entity ranges from 70% to 90%.[7] The longer a mention is, the more information it represents, e.g., *a person* vs. *a person that works at Penn*. Overall, the first mention captures semantic meaning of an entity.

**Semantic Similarity.** In addition to the first mention, an entity can accumulate information with each subsequent mention. The mentions can be identical, slightly different, or completely different when compared to other mentions within the same entity. To examine the semantic similarity of coreferent mentions, we compute the Euclidean distance between the embeddings of each gold mention pair encoded using mBERT (Devlin et al., 2019).

In Figure 3, a greater distance indicates that the mentions have a bigger semantic distance, while still referring to the same entity. Conversely, a smaller distance suggests that the mentions are semantically more similar, if not identical. We speculate that the genres of the datasets have an

---

[6]See Appendix A.1 for the details of proper noun and zero pronoun.

[7]See Appendix A.2 for the details.

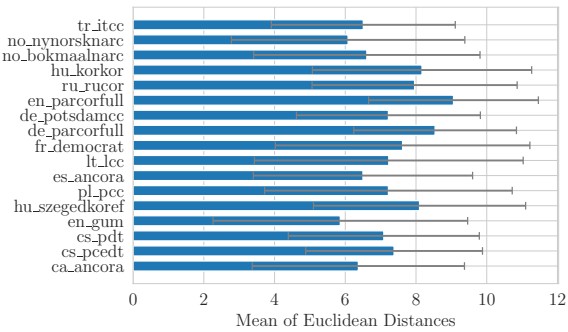

Figure 3: Mean and variance of Euclidean distances between pairs of mentions within the same entities across all datasets.

impact on the analysis above. For example, in narrative texts such as EU Bookshop publications in EN_PARCORFULL (Lapshinova-Koltunski et al., 2018) and Hungarian Wikipedia in HU_KORKOR (Vadász, 2020; Vadász, 2022), an entity can be realized with different expressions. Thus, the semantic similarity of mentions tends to be greater. Recall that nominal noun and proper noun are two main categories of mention types. So, it is challenging to resolve mentions that have bigger semantic distance.

### 3.3 Document

In a document, there can be multiple entities, with some entities spanning the entire document while others appearing only in very few adjacent sentences. Occasionally, these entities may overlap within certain sections of the document, particularly in areas where complex relationships between entities are discussed. Table 3 shows an example text.

**Competing Antecedents of Pronominal Anaphors.** In a local context, the resolution of pronouns can become difficult due to their

---

The study of how [people]₈, as [fans]₈, access and manage information within a transmedia system provides valuable insight that contributes not only to [practitioners]₇ and [scholars of the media industry]₆, but to the wider context of cultural studies, by offering findings on this new model of [the fan]₅ as [consumer]₄ and [information-user]₃. For [us]₁, as [digital humanists]₁, defining [the "transmedia fan"]₂ is of particular relevance as [we]₁ seek to understand contemporary social and cultural transformations engendered by digital technologies.

Table 3: Text with indexed entities in the two adjacent sentences. All mentions in bold are valid competing antecedents of the pronominal anaphor in blue.

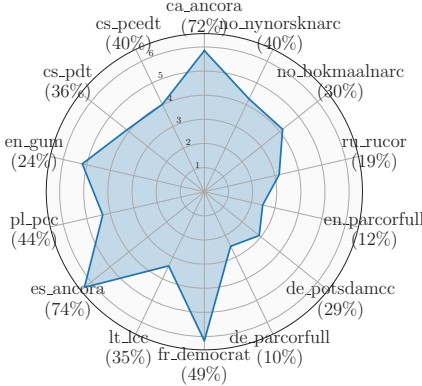

Figure 4: Average number of competing antecedents for pronominal anaphors. The percentages next to the datasets represent the percentage of valid examinations of overt pronouns.

ambiguity caused by the presence of multiple potential antecedents from distinct entities or singletons. We focus on those ambiguous cases that have potential antecedents with gender and number agreement. Both the pronouns and their antecedents are located in the same or the immediately preceding sentence.

Figure 4 shows that in CA_ANCORA and ES_ANCORA (Recasens and Martí, 2010), over 70% of overt pronouns satisfy the analysis conditions mentioned in the previous paragraph. This percentage is notably higher compared to the other datasets. Additionally, the average number of competing candidates in these two datasets is around six. This highlights the considerable difficulty in distinguishing the true antecedent(s) of the pronoun among a pool of antecedents. To address such complex scenarios, one heuristic and explainable approach is to leverage centering theory (Grosz et al., 1995). It suggests that a pronoun tends to refer to the center or the most prominent entity in the preceding context. Specifically, by tracking the center transitions, we can identify potential antecedents based on salience and continuity of the entity. Centering theory is applicable across all languages, as it is not dependent on any specific language.

Besides analyzing overt pronouns, we also examine the competing antecedents for zero pronouns. In the Czech datasets (Nedoluzhko et al., 2016; Hajič et al., 2020), the average number of competing antecedents is less than four, which is lower than that of CA_ANCORA and ES_ANCORA.[8] This implies that identifying the true antecedents of zero

---

[8]See Appendix A.3 for the details.

anaphors is not very difficult in the Czech datasets. In pro-drop languages, a more coherent discourse tends to facilitate or encourage the use of zero pronoun especially in dialogue or social media contexts. We found that the nearest antecedents of some zero pronouns can either be overt pronouns or zero pronouns that are less informative. Hence, resolving anaphoric zero pronouns is a difficult subtask that requires contextual information.

### 3.4 Genre

A document can be different in types of discourse with respect to referring expressions. For example, authors may use diverse expressions (e.g., *dog owners, owners, puppy owners* and *they*) when referring to the same entity for the physical continuity of the text. In contrast, spoken discourse, especially in conversations, tends to have a higher density of referring expressions, including many pronouns and ellipsis, which contribute to the grammatical coherence within the discourse (e.g., *Sue? Is not here.*), and relies mostly on shared situational knowledge between speaker and listener (known as the 'common ground'). (Dorgeloh and Wanner, 2022)

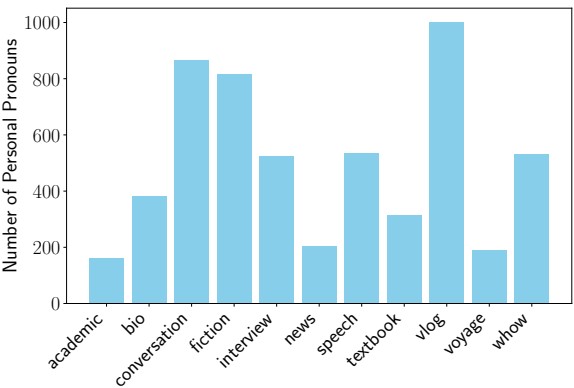

Figure 5: Number of personal pronouns per 8000 words in various genres within EN_GUM (Zeldes, 2017).

In Figure 5, we present the frequency of personal pronouns usage per eight thousands words in each genres from the English corpus, EN_GUM (Zeldes, 2017). The results show that **vlog**, as a type of web discourse, has the highest frequency of pronoun usage. Different from **conversation**, content creators record themselves on video for their audience without engaging in real-time interaction during the recording process. When they share their thoughts or experiences, they tend to use first-person pronouns (e.g., *I* and *we*) more frequently compared to other genres. We also observe that the frequency of pronouns in **fiction** is high, surpassing even that of

**speech**, indicating a strong continuity in reference, particularly related to the story's characters. This finding is in line with the results of Dorgeloh and Wanner (2022). As for written non-fiction, particularly in **academic**, **news** and **voyage** (describing a journey or trip), there is a lower use of pronouns, with academic texts showing the lowest frequency.

## 4 Error Analysis of MCR Systems

Apart from studying coreference on gold annotations solely, we also investigate the ground truth that the MCR systems failed to address. Our particular focus is on two-mention entities, which comprise over 80% of the gold entities where the recall is zero.[9] Here, we analyze the predictions of two MCR systems: BASELINE (Pražák et al., 2021), an end-to-end based system, and ÚFAL (Straka and Straková, 2022), the winning system in the CRAC 2022 shared task on MCR.[10] Figure 6 presents the error analysis in a tree structure conducted on ÚFAL.

### 4.1 Undetected Mentions

The primary factor leading to unresolved two-mention entities is the inability to detect one or both of the mentions. ÚFAL identifies 22% of the mentions, while BASELINE detects 19%. ÚFAL employs a pipeline approach, treating mention detection as a separate token-level classification task. The proposed tags for tokens can handle embedded and also overlapping mention spans. We speculate that the mention detection module contributes slightly more to the identification of mentions.

We further analyze the mention types and length of the undetected mentions.[9] **(i)** More than 50% of the undetected mentions on average are nominal nouns, so we try to analyze the types of these noun phrases based on definiteness, such as demonstrative articles (e.g., *that house*) and proper noun-modified noun phrases (e.g., *Barack Obama presidency*). However, due to the highly variable nature of definiteness across languages and the lack of consistent annotations at this level of granularity, we encounter a challenge in implementing this analysis. For example, in Lithuanian, definiteness is encoded within adjectives or nouns, and possessive adjectives in Hungarian can only be inferred from word suffixes. Moreover, some languages, such

---

[9]See Appendix B for the details of the error analysis.

[10]The two system outputs from the development sets of CorefUD 1.0 are publicly accessible at https://ufal.mff.cuni.cz/corefud/crac22.

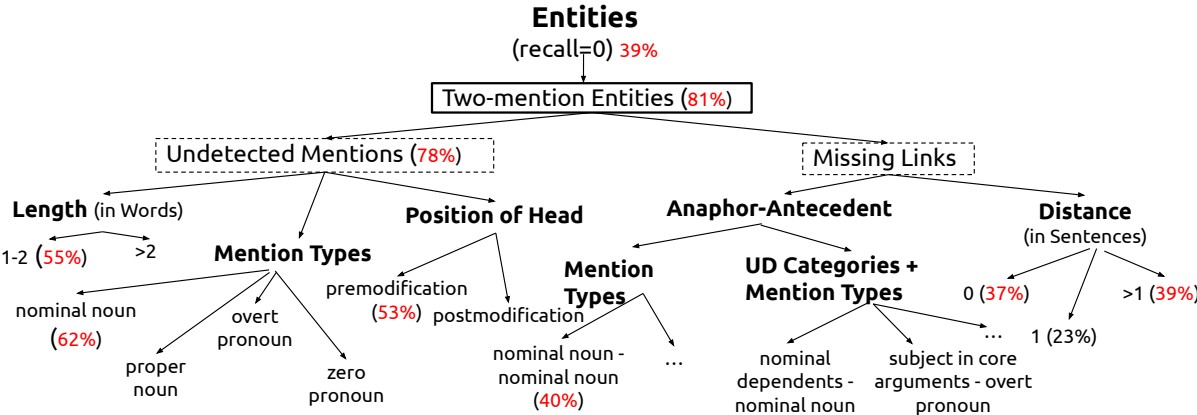

Figure 6: Error analysis of entities where ÚFAL fails to resolve, meaning that the recall of these entities is zero. For example, 81% of unresolved entities consist of two mentions. One of the reasons for the failure to resolve two-mention entities is that 78% of the mentions within those entities are not detected. The figures are computed on average across all datasets.

as Slavic ones, do not have grammaticalized definiteness at all. **(ii)** Analyzing mention length, we observe that the majority of mentions in Hungarian (70%) and Lithuanian (80%) consist of only one or two words. One of the reasons is that Hungarian, for example, is an agglutinative language[11]. When dealing with such languages, it is plausible to include a preprocessing stage to handle word splitting.

## 4.2 Missing Links

We also explore the relationship between the two mentions in the unresolved entities.[9] First, we notice that in BASELINE, more than 45% of the entities have both mentions located in the same sentence. To resolve those entities, syntax information that captures the grammatical relationships and dependencies between words within the sentences is beneficial. One approach is employing binding theory (Chomsky, 1993). On the other hand, in ÚFAL, 39% of the entities have their two mentions spanning across multiple sentences. To address this issue, an approach is to use knowledge extracted from the discourse structure of the text. Second, for both systems, resolving cases where both mentions are nominal nouns presents difficulties across all languages. Additionally, our analysis in Section 3 demonstrates that there are mention pairs referring to the same entities, but showing lower semantic similarity. These findings suggest that it

is important to improve the capability of resolving noun phrases. Lastly, we examine the gold anaphor-antecedent relations between the two mentions of the unresolved entities. We found that the most frequent UD relation associated with the antecedents of nominal nouns are nominal dependents (e.g., nominal modifier and appositional modifier). For antecedents of overt pronouns, the subject in core arguments is the most common UD relation.

## 5 Modeling with Universal Annotations

Based on the findings above, we can gain additional insights and clues regarding MCR. For example, we found that the closest antecedents of overt pronouns are always located in subject position. This pattern is common in nearly all languages as shown in Figure 2 (b). Therefore, we use linguistic information extracted from universal annotations for the purpose of modeling and examine its effectiveness, in the following section.

### 5.1 Model

**Baseline.** We adopt the model proposed by Pražák et al. (2021) as our base model, which is an end-to-end neural model inspired by the method introduced by Lee et al. (2017). It serves as the baseline for the CRAC 2022 shared task on multilingual coreference resolution.

**Incorporating Linguistic Information.** Given an input document consisting of $n$ tokens, we first generate a contextual embedding for each token using mBERT denoted as $\mathbf{X} = (\mathbf{x}_1, ..., \mathbf{x}_n)$. The tokenization is based on either word forms (**wf**) or

---

[11]Words are constructed by combining stem forms with multiple affixes to convey diverse grammatical features such as tense and number, for example, *beleselkedtem* (*I look into*) and *Odafigyelhettél volna* (*You could have paid attention to it*).

| MODELS | AVG | CA | CS PCED | CS PDT | EN GUM | HU | PL | ES | LT | FR | DE PARC | DE POTS | EN PARC | RU |
|---|---|---|---|---|---|---|---|---|---|---|---|---|---|---|
| BASELINE | 53.7 | 55.2 | 68.4 | 64.3 | 48.8 | 46.4 | 50.2 | 57.6 | **64.2** | 57.0 | 33.7 | 43.3 | 43.0 | **66.9** |
| ours_lem | 51.7 | 52.8 | 65.0 | 62.7 | 48.1 | 44.0 | 44.3 | 54.6 | 60.2 | 56.7 | 30.6 | 40.8 | 48.4 | 64.5 |
| ours_wf | **54.6** | **55.7** | **68.5** | **64.9** | **50.1** | **47.1** | **50.4** | **57.7** | 62.1 | **58.6** | **35.1** | **44.9** | **48.5** | 66.5 |
| ⊖ ua | -0.51 | -0.03 | -0.23 | 0.00 | -0.75 | -0.21 | -0.72 | +0.34 | +1.57 | -1.78 | +0.81 | -2.90 | -4.04 | +1.36 |
| ⊖ lang | -0.87 | -0.45 | -0.11 | -0.65 | -1.29 | -0.71 | -0.19 | -0.14 | +2.09 | -1.62 | -1.44 | -1.61 | -5.58 | +0.40 |

Table 4: F1 scores on the test set in our setting are reported on average across three runs. ⊖ rules out the additional features extracted from universal annotations (ua) and languages (lang) from **ours_wf** for the ablation study.

lemmas (**lem**). Then we define the embedding of each candidate span $c$ as:

$$\mathbf{e}_c = [\mathbf{x}_{c_{start}}, \mathbf{x}_{c_{end}}, \hat{\mathbf{x}}_c, \phi(s_c)]$$

where $\mathbf{x}_{c_{start}}$ and $\mathbf{x}_{c_{end}}$ denote the embeddings of the boundary tokens. $\hat{\mathbf{x}}_c$ is the addition of attentionally weighted token representations in the candidate. $\phi(s_c)$ is a concatenated feature vector that includes the width, UPOS tags, UD relations, mention types and UD categories of the span. We select the token with the maximum attention weight as the head of the candidate to compute the mention types and UD categories as discussed in Section 3.

We measure how likely a candidate is a mention by using a mention score $f_m(\cdot)$:

$$f_m(c) = \mathbf{FFNN}_m([\mathbf{e}_c, \phi(u_c)])$$

where $\phi(u_c)$ encodes the UPOS tag, UD relation, mention type and UD category of the candidate determined by its 'head' word as mentioned above.

After extracting the top $\lambda n$ mentions based on the mention score, we compute the likelihood of a candidate mention $c$ being an antecedent of a query mention $q$ by a scoring function $f(c, q)$:

$$f(c, q) = \mathbf{FFNN}_s([\mathbf{e}_c, \mathbf{e}_q, \mathbf{e}_c \circ \mathbf{e}_q, \phi(c, q)])$$

$\phi(c, q)$ denotes the embeddings of some general features of the document: language and word order of the language[12]. For each query mention, our model predicts a distribution $\hat{P}(q)$ over its candidates, $q \in Y(c)$:

$$\hat{P}(q) = \frac{\exp(f(c, q))}{\sum_{k \in Y(c)} \exp(f(c, k))}$$

Note that if the query mention is a singleton, we set the scoring function to zero.[13]

---

[12] https://wals.info/

[13] For more details, please refer to the original papers, Pražák et al. (2021) and Lee et al. (2017).

**Training and Inference.** Since ÚFAL (Straka and Straková, 2022) demonstrated that a multilingual model based on a multilingual language model outperforms monolingual models on the MCR task, we adopt a similar approach. Our model is jointly trained on a mixture of datasets of 10 languages from CorefUD 1.0 (Nedoluzhko et al., 2022) using mBERT (Devlin et al., 2019) as the pretrained language model. Then we use this trained model to predict mention clusters on the target language-specific datasets.

## 5.2 Experiments

**Settings.** We verify the effectiveness of our models on CorefUD 1.0 (Nedoluzhko et al., 2022). Because the test datasets are not publicly available, we partitioned approximately 10% of the training datasets to create our own test datasets. The results are reported using the CoNLL F1 score — the average of MUC (Vilain et al., 1995), B3 (Bagga and Baldwin, 1998), CEAFe (Luo, 2005). The final ranking score is calculated by macro-averaging the CoNLL F1 scores over all datasets. To ensure a fair comparison, we keep all parameters the same as the baseline (Pražák et al., 2021). All our experiments are performed on a single NVIDIA Tesla V100 32G GPU. We examine two models, namely **ours_wf** and **ours_lem**, as discussed in Section 5.1, in comparison with the baseline model trained on our specific setting.

**Results.** Table 4 presents our results. Our model **ours_wf** shows a modest improvement over the baseline with a margin of 0.9% F1 score on average across all languages. The model performs best on Germanic datasets, whereas the LT_LCC (Žitkus and Butkiene, 2018) and RU_RUCOR (Toldova et al., 2014) datasets present the greatest difficulties, indicating that these two Baltic and Slavic languages are particularly difficult to handle. In the ablation study, we observe that including general features like language and word order also yields

positive effects on performance, in addition to incorporating universal annotations.

In contrast, the performance of **ours_lem** shows a decline compared with BASELINE. The method is specifically designed to address data sparsity and handling out-of-vocabulary words in morphological-rich languages. However, lemmatization can result in different words being mapped to the same lemma and loss of valuable morphological information present in word forms. In order to handle multiple languages together, it is crucial to employ a trade-off strategy or to implement a preprocessing approach.

**Error Analyses.** We employ the same analysis methodology as presented in Section 4 for the error analysis of our model **ours_wf** and BASELINE *in our setting*. We found that **ours_wf** predicts more clusters correctly than BASELINE, either in full or partially (i.e., the rate of gold entities with a recall of zero is lower on average, 39.19% vs. 39.77%.). Two-mention entities are the most difficult cases for the two examined systems. In these unresolved two-mention entities, **ours_wf** has fewer undetected mentions on average especially in FR_DEMOCRAT and DE_PARCORFULL, as illustrated in Figure 7. Among those undetected, there are more mentions consisting of more than two words compared with BASELINE. For the missing links, the two systems produce similar results. Both mentions in two-mention entities are primarily nominal nouns. And the most frequent UD relation associated with the antecedent of nominal is still nominal dependents. Overall, our model **ours_wf** can resolve slightly more entities and shows a superior performance in mention detection compared with BASELINE. Nevertheless, there is still room for improvement.

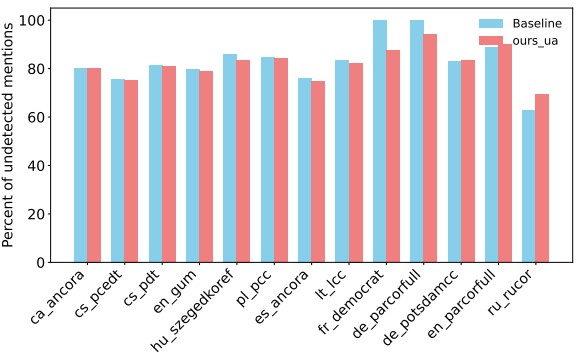

Figure 7: Percentage of undetected mentions in unresolved two-mention entities.

## 6 Discussion and Conclusion

It has become apparent that leveraging universal morphosyntactic annotations can be advantageous in various ways, like exploring underlying patterns of coreference, performing in-depth analysis and making a contribution to the development of an MCR system. However, there are still language-specific characteristics that hinder the comprehensive study of multiple languages together, particularly when it involves analyzing intricate aspects of the morphological layer, like definiteness and compound nouns in German. In addition, while multilingual datasets are harmonized to some extent, there are still cases where certain information, such as entity types, is only provided for a limited number of languages. This limitation prevents us from conducting further analyses, such as examining semantic class agreement across languages. We study MCR primarily focusing on identity coreference since it is the most important relation across all datasets. However, it is important to note that there exist various other anaphoric relations, such as bridging and discourse deixis (Yu et al., 2022), that remain unexplored. In this work, we analyze coreference across multiple languages by leveraging the harmonized universal morphosyntactic and coreference annotations in CorefUD. This analysis provides valuable insights into common features and challenges in MCR. We demonstrate the benefits of incorporating linguistic features for enhancing the MCR system performance.

## Limitations

In this work, our analyses are mainly corpus-based studies. The reliance on selected specific corpora may result in a focus on particular genres, domains, or time periods that may not be representative of other contexts. However, with the high number of datasets from diverse genres and domains, we believe the findings still can provide some valuable insights into MCR. The languages examined in our study belong to the European language group. It would be interesting to involve languages from other regions, like Arabic and Chinese.

## Acknowledgements

We thank the anonymous reviewers for their helpful feedback that greatly improved the final version of the paper. We also thank Margareta Kulcsar for her early experiments contributing to this work.

This work has been funded by the Klaus Tschira Foundation, Heidelberg, Germany. The first author has been supported by a HITS Ph.D. scholarship.

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

# A  Linguistic Analyses on CorefUD 1.1

We present the statistics of CorefUD 1.1 in Table 5 to provide a basic understanding of all the datasets.

## A.1  Anaphor-Antecedent Relation

Figure 8 demonstrates the analysis of anaphor-antecedent relations where anaphors are proper noun and zero pronoun.

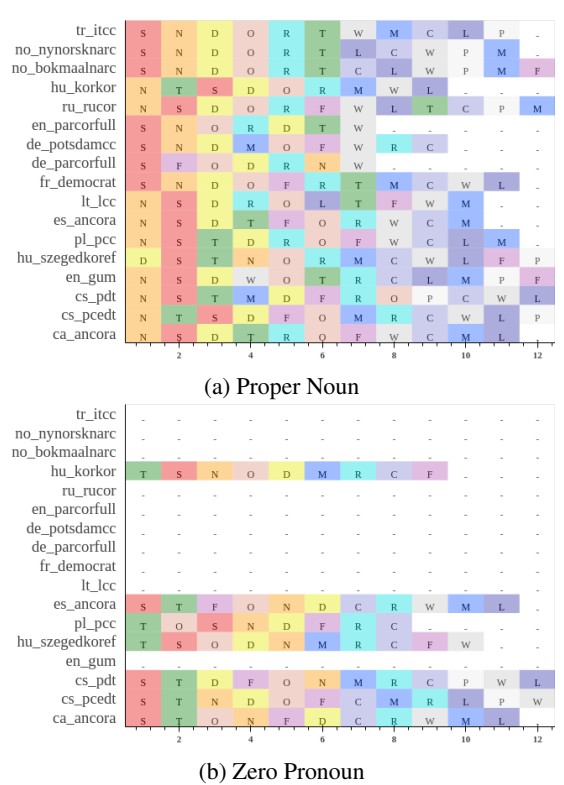

(a) Proper Noun

(b) Zero Pronoun

Figure 8: The figure presents the ranking of relations between UD categories of antecedents and mention types of anaphors in each dataset. The mention types include proper noun (a) and zero pronoun (b). See Table 2 for the full name of UD categories.

## A.2  First Mention

Table 6 shows the statistics of first mentions that are the longest mentions in entities.

## A.3  Competing Antecedents of Pronominal Anaphors

Figure 9 shows the analysis of competing antecedents of zero pronouns on three languages, Catalan, Czech and Spanish.

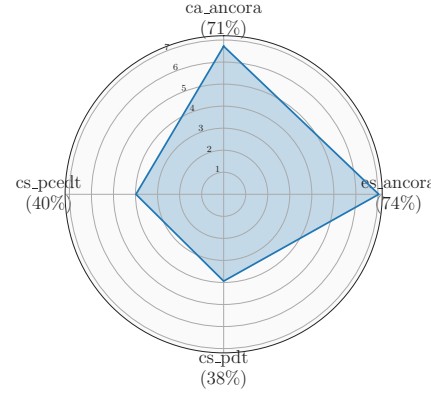

Figure 9: The figure displays the average number of competing antecedents for zero pronouns. The percentages indicated next to the datasets represent the percentage of valid examinations of zero pronouns.

# B  Error Analysis

Figure 10 presents the percentages of mention types of undetected mentions based on the predictions of ÚFAL.

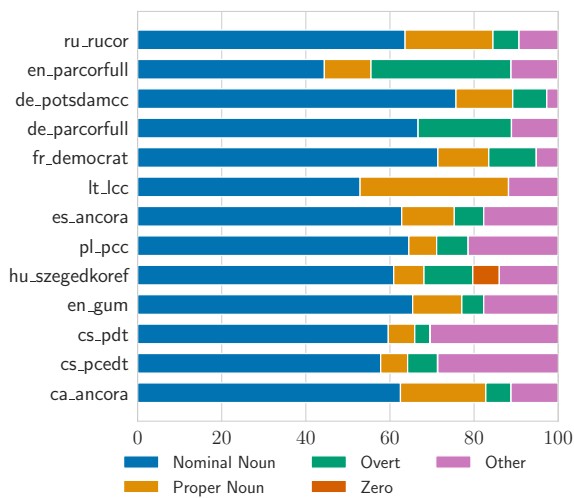

Figure 10: Coverage of mention types in the undetected mentions divided in percentage for ÚFAL.

In Figure 11, we show the distances between mentions in the unresolved two-mention entities for BASELINE and ÚFAL.

Table 7 shows various analyses conducted to explore the underlying reasons of unresolved entities in both BASELINE and ÚFAL systems.

| | Docs | Sents/Doc | Tokens/Sent | Entities | Mentions | Mentions/Entity |
|---|---|---|---|---|---|---|
| CA_ANCORA | 1,011 | 10.52 | 30.39 | 13,589 | 48,323 | 3.56 |
| CS_PCEDT | 1,875 | 21.24 | 23.42 | 39,945 | 136,932 | 3.43 |
| CS_PDT | 2,533 | 15.29 | 16.85 | 36,378 | 120,024 | 3.30 |
| EN_GUM | 151 | 56.61 | 17.05 | 5,816 | 25,615 | 4.40 |
| HU_SZEGEDKOREF | 320 | 22.31 | 14.08 | 3,862 | 12,278 | 3.18 |
| PL_PCC | 1,463 | 19.63 | 15.03 | 17,748 | 65,915 | 3.71 |
| ES_ANCORA | 1,080 | 10.50 | 31.63 | 15,532 | 56,668 | 3.65 |
| LT_LCC | 80 | 16.63 | 22.62 | 879 | 3,609 | 4.11 |
| FR_DEMOCRAT | 50 | 207.64 | 21.58 | 5,718 | 38,490 | 6.73 |
| DE_PARCORFULL | 15 | 30.47 | 18.93 | 192 | 737 | 3.84 |
| DE_POTSDAMCC | 142 | 12.80 | 14.68 | 715 | 2,027 | 2.83 |
| EN_PARCORFULL | 15 | 30.47 | 19.18 | 158 | 710 | 4.49 |
| RU_RUCOR | 145 | 48.75 | 17.48 | 2,803 | 12,509 | 4.46 |
| HU_KORKOR | 76 | 14.29 | 17.91 | 874 | 3,178 | 3.64 |
| NO_BOKMAALNARC | 284 | 46.02 | 15.55 | 4,647 | 21,828 | 4.70 |
| NO_NYNORSKNARC | 336 | 30.71 | 16.74 | 4,307 | 18,354 | 4.26 |
| TR_ITCC | 19 | 185.89 | 12.46 | 523 | 2,602 | 4.98 |

Table 5: Statistics of CorefUD 1.1 based on the number of documents (Docs), average number of sentences per document (Sents/Doc), average number of tokens per sentence (Tokens/Sent), number of entities, number of mentions, and average number mentions per entity (Mentions/Entity) in each of the datasets.

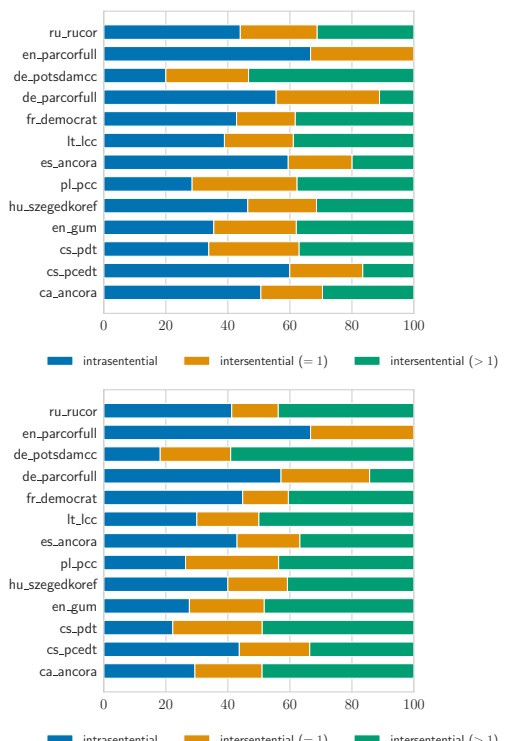

Figure 11: Coverage of distance between mentions in the unresolved two-mention entities divided in percentage. The top figure is for BASELINE, and the bottom one corresponds to ÚFAL.

| | Entities (%) |
|---|---|
| CA_ANCORA | 76.92 |
| CS_PCEDT | 87.10 |
| CS_PDT | 75.22 |
| EN_GUM | 69.55 |
| HU_SZEGEDKOREF | 80.11 |
| PL_PCC | 83.13 |
| ES_ANCORA | 77.25 |
| LT_LCC | 81.80 |
| FR_DEMOCRAT | 82.46 |
| DE_PARCORFULL | 88.02 |
| DE_POTSDAMCC | 77.48 |
| EN_PARCORFULL | 89.24 |
| RU_RUCOR | 83.41 |
| HU_KORKOR | 82.27 |
| NO_BOKMAALNARC | 81.62 |
| NO_NYNORSKNARC | 78.11 |
| TR_ITCC | 70.36 |

Table 6: Percentage of first mentions that are the longest mentions within entities in the corresponding datasets.

| | A | | B | | C | | D | | E | | F | |
|---|---|---|---|---|---|---|---|---|---|---|---|---|
| | BL | ÚFAL | BL | ÚFAL | BL | ÚFAL | BL | ÚFAL | BL | ÚFAL | BL | ÚFAL |
| CA_ANCORA | 34.14 | 21.06 | 81.30 | 82.40 | 80.71 | 81.90 | 36.20 | 40.30 | 64.30 | 62.5 | 8.56 | 6.91 |
| CS_PCEDT | 33.80 | 20.71 | 88.15 | 88.49 | 77.74 | 68.83 | 41.60 | 36.90 | 53.70 | 59.5 | 7.74 | 6.35 |
| CS_PDT | 35.50 | 23.56 | 84.70 | 88.17 | 82.78 | 79.22 | 46.60 | 48.80 | 54.90 | 55.2 | 5.87 | 4.62 |
| EN_GUM | 38.85 | 25.04 | 75.19 | 83.33 | 86.45 | 79.66 | 48.30 | 50.60 | 61.30 | 65.4 | 4.87 | 4.83 |
| HU_SZEGEDKOREF | 39.23 | 34.01 | 80.92 | 83.33 | 78.21 | 82.80 | 69.70 | 70.20 | 56.60 | 53.1 | 2.38 | 2.32 |
| PL_PCC | 36.97 | 23.05 | 85.78 | 90.25 | 83.07 | 82.18 | 61.20 | 61.00 | 31.00 | 31.8 | 4.05 | 4.06 |
| ES_ANCORA | 35.50 | 16.33 | 85.29 | 90.37 | 81.32 | 78.52 | 32.00 | 37.20 | 68.20 | 66.3 | 9.29 | 7.78 |
| LT_LCC | 27.84 | 15.46 | 66.67 | 66.67 | 88.89 | 85.00 | 87.50 | 70.60 | 15.60 | 29.4 | 1.47 | 2.00 |
| FR_DEMOCRAT | 43.22 | 25.20 | 76.18 | 87.63 | 85.19 | 76.07 | 52.80 | 56.30 | 61.10 | 57.3 | 3.62 | 3.61 |
| DE_PARCORFULL | 45.45 | 36.36 | 90.00 | 87.50 | 72.22 | 64.29 | 53.80 | 66.70 | 46.20 | 55.6 | 3.54 | 2.67 |
| DE_POTSDAMCC | 50.59 | 27.06 | 69.77 | 95.65 | 98.33 | 84.09 | 47.50 | 51.40 | 74.60 | 75.7 | 3.58 | 3.87 |
| EN_PARCORFULL | 43.75 | 37.50 | 85.71 | 100.00 | 66.67 | 75.00 | 37.50 | 55.60 | 62.50 | 44.4 | 6.25 | 5.44 |
| RU_RUCOR | 35.80 | 22.17 | 80.65 | 83.33 | 74.80 | 80.63 | 61.80 | 67.90 | 32.60 | 34.9 | 2.66 | 2.41 |

Table 7: The table shows the results of the error analysis on BASELINE (BL) and ÚFAL across all datasets in CorefUD 1.0. All results are presented as percentages excluding F. The initials for the sub analyses stand for: **A**: Unresolved entities (recall=0). **B**: Two-mention entities in the unresolved entities. **C**: Udetected mentions in the unresolved two-mention entities. **D**: Mentions that only have one or two words in the undetected mentions. **E**: Undetected mentions that are pre-modifications. **F**: Average length of mentions that are not detected.