# OpenReview forum: "Investigating Multilingual Coreference Resolution by Universal Annotations"
_EMNLP/2023/Conference — EMNLP 2023 Findings_

### Official Review · Reviewer_eyYv · 2023-08-05

**Soundness:** 4

**Excitement:**

3: Ambivalent: It has merits (e.g., it reports state-of-the-art results, the idea is nice), but there are key weaknesses (e.g., it describes incremental work), and it can significantly benefit from another round of revision. However, I won't object to accepting it if my co-reviewers champion it.

**Missing References:**

0.9 F1 -> 0.9% F1

**Paper Topic And Main Contributions:**

This paper presents an in-depth multilingual description of CorefUD 1.1, highlighting various coreference properties differences in the 12 languages represented in this corpus. Then, a proposed model, which concatenates the UPOS tag, UD relation, mention type and UD category of the candidate, is proposed. This model is compared with the baseline model used in the shared task of multilingual coreference resolution (CRAC 2022) using CorefUD 1.0 (10 languages). Finally, the proposed model outperforms the baseline by 0.9% of F1, and an ablation test is presented.

**Questions For The Authors:**

In Table 4, what is the different of between the baseline model and the proposed one without UA and LANG? Specifically, what is the characteristic of the model that accounts for 0.03?

**Reasons To Accept:**

The multilingual in-depth description of CorefUD 1.1

The hybrid coreference model, which combines mBERT with morphosyntactic information.

**Reasons To Reject:**

The statistical significance of the results is not addressed.

No error analysis is presented. (During the rebuttal phase, the authors indicate that they may put it into the final version, but it is unclear what analysis will be done).

**Reproducibility:**

4: Could mostly reproduce the results, but there may be some variation because of sample variance or minor variations in their interpretation of the protocol or method.

**Reviewer Confidence:**

4: Quite sure. I tried to check the important points carefully. It's unlikely, though conceivable, that I missed something that should affect my ratings.

---

> ### Author Rebuttal · Authors · 2023-08-28
>
> Thanks for reviewing our paper and your valuable feedback.
>
> > The statistical significance of the results is not addressed. It is also unclear whether the results are from a single run or the average of repeated runs.
>
> We follow the standard practice in the coreference resolution literature, and repeat our experiments three times to ensure the reliability of our results. We report the mean and standard deviation of F1 scores in our setting, as shown in the table below. We note that the models we investigated exhibited relatively large standard deviations across datasets. We attribute this variance gap to the differences in dataset sizes, as the models appear to yield higher variances when evaluated on a small corpus such as DE-PARC and EN-PARC.
>
> | Models| CA|CS-PCED| CS-PDT|EN-GUM|HU|PL|ES|LT|FR|DE-PARC|DE-POTS|EN-PARC|RU|
> |-|-|-|-|-|-|-|-|-|-|-|-|-|-|
> |Baseline|55.2±0.6|68.4±0.5|64.3±0.4|48.8±1.2|46.4±1.0|50.2±0.5|57.6±0.5|64.2±1.5|57.0±0.7|33.7±1.7|43.3±1.4|43.0±1.5| 66.9±1.1|
> |ours_lem|52.8±0.5|65.0±0.4|62.7±0.3|48.1±1.0|44.0±1.1|44.3±0.6|54.6±0.4|60.2±1.7|56.7±0.8|30.6±1.5|40.8±1.5|48.4±1.8|64.5±0.9|
> |ours_wf|55.7±0.5|68.5±0.4|64.9±0.5|50.1±1.1|47.1±0.9|50.4±0.5|57.7±0.6|62.1±1.5|58.6±0.8|35.1±1.8|44.9±1.4|48.5±1.8|66.5±1.1|
>
> > No error analysis is presented.
>
> We will include this error analysis following the method outlined in section 4 in the final version where we will have one more page available.
>
> > The claim of 0.9 F1 improvement is somehow misleading. The F1 ranges from 0 to 1. Thus, without clear context, it looks impressive, but it is because the authors use a range between 0 to 100. (0.9 F1 -> 0.9% F1)
>
> We will use 0.9% F1 in the final version.
>
> > In Table 4, what is the different of between the baseline model and the proposed one without UA and LANG? Specifically, what is the characteristic of the model that accounts for 0.03?
>
> We are sorry but we do not understand what you are referring to with 0.03. Please clarify during the discussion period.

---

### Official Review · Reviewer_5r1p · 2023-08-05

**Soundness:** 4

**Excitement:**

4: Strong: This paper deepens the understanding of some phenomenon or lowers the barriers to an existing research direction.

**Paper Topic And Main Contributions:**

The paper presents an application of the data sets from CorefUD. The goal is to understand multilingual coreference resolution and to provide a solution for this.

**Reasons To Accept:**

The paper focuses on linguistic properties of mentions, entities and documents and analyzes the data to avoid frequent coreference anotation errors. It is very important to have data sets checked by human annotators.

**Reasons To Reject:**

The paper do not explain the notions presented in the paper such as Root Cause Analysis or the method to find undetected mentions, the antecedents.

**Reproducibility:**

4: Could mostly reproduce the results, but there may be some variation because of sample variance or minor variations in their interpretation of the protocol or method.

**Reviewer Confidence:**

4: Quite sure. I tried to check the important points carefully. It's unlikely, though conceivable, that I missed something that should affect my ratings.

---

> ### Author Rebuttal · Authors · 2023-08-28
>
> Thanks for reviewing our paper and your valuable feedback.
>
> > The paper do not explain the notions presented in the paper such as Root Cause Analysis or the method to find undetected mentions, the antecedents.
>
> In lines 60-67, we provide an explanation of root cause analysis. We will substitute the term with 'error analysis' for easier comprehension. We employ the evaluation metric LEA (Moosavi and Strube, 2016) to identify unresolved clusters (recall=0) and find the undetected mentions by comparing the gold mentions with the predicted mentions generated by MCR systems.

---

### Official Review · Reviewer_qVbW · 2023-08-06

**Soundness:** 3

**Excitement:**

3: Ambivalent: It has merits (e.g., it reports state-of-the-art results, the idea is nice), but there are key weaknesses (e.g., it describes incremental work), and it can significantly benefit from another round of revision. However, I won't object to accepting it if my co-reviewers champion it.

**Paper Topic And Main Contributions:**

The article presents three contributions: One is a quantitative analysis of the data in the CorefUD 1.1 coreference corpus, the second one is an error analysis of the coreference resolution system of Straka and Strakova (2022), and the third is the authors' own coreference system based on the baseline system of Prayak et al (2021) where the authors add a small number of additional features.
For the first part, the authors look at mention internal structure, the coreference chains ("entities") themselves, and finally document-level information such as the number of competing antecedent candidates. Looking at both papers that mainly concern themesleves with (prior or current versions of) the CorefUD corpus - Nedoluzhko et al 2022 and Zabortsky et al 2022, the quantitative analysis seems to be novel with respect to these works which do not touch the content of the coreference corpora at all.
The quantitative analysis looks at properties of each of the language-specific subcorpora of CorefUD; in doing so, no provision is made to tease apart confounders such as genre/domain of the texts, differences in annotation styles (even within a unified scheme - this was found in UD as a significant source of variation) etc., but would still be somewhat useful to researchers implementing their own coreference systems or preparing deeper analyses.

**Questions For The Authors:**

(A) Why didn't you include your own system in the error analysis, or the system of Straka and Strakova in the Sec. 6 comparison?

**Reasons To Accept:**

The paper produces several angles of analysis which together help us learn more about the CorefUD corpus than was available before

**Reasons To Reject:**

Leaving out the common theme of using the CorefUD corpus, the three parts of this article are pretty much independent of each other rather than building on a common technical contribution - in particular the system building experiment in section 5 does not in any way profit from the corpus analysis or error analysis sections. If one wanted to argue against this paper, it is that neither of the sections has the contribution height for being accepted as a main conference paper - the quantitative analysis looks like it stays at the level of what could be done in short time by taking at face value the corpus annotations rather than looking more deeply into possible confounders (and I believe the authors if they say this part has gotten substantial attention and rounds of refinement - in UD it took a village of experts several releases to unearth cross-language inconsistencies); The error analysis in section 4 states relatively obvious facts that are going to affect any SotA coreference system, while this is a truism for error analyses this is expected to benefit from the cross-system comparison with their own system that the authors mention in the rebuttal.
In the related works presentation, the classification of MCR systems seems a bit confused - it presents the rule-based approach of Lee et al 2011 as "Template-based" which I don't find very helpful as a term, and in the subsequent text fails to cleanly separate multilingual (training data in multiple languages) from cross-lingual (training data in one language and testing in another language) tasks or even mixes training methodology (latent structure, joint training) and features (methods with linguistic information).

**Reproducibility:**

4: Could mostly reproduce the results, but there may be some variation because of sample variance or minor variations in their interpretation of the protocol or method.

**Reviewer Confidence:**

4: Quite sure. I tried to check the important points carefully. It's unlikely, though conceivable, that I missed something that should affect my ratings.

**Typos Grammar Style And Presentation Improvements:**

l. 088 - The term two-mention entities isn't clear in context
l. 130 - morphologically rich

---

> ### Author Rebuttal · Authors · 2023-08-28
>
> Thanks for reviewing our paper and your valuable feedback.
> > no provision is made to tease apart confounders such as genre/domain of the texts, differences in annotation styles (even within a unified scheme - this was found in UD as a significant source of variation) etc.
>
> We will conduct an additional genre/domain analysis on a few datasets, such as English-Gum (Zeldes, 2017), as only a limited number of datasets have genre/domain annotations in CorefUD. As for the differences in annotation styles, our work mainly focuses on analyzing the linguistic phenomenon of coreference, based on the multilingual coreference dataset proposed by Nedoluzhko et al. (2022). In Nedoluzhko et al. (2022), the authors have already discussed a lot about the diverse annotation schemes employed by all teams.
>
> > Leaving out the common theme of using the CorefUD corpus, the three parts of this article are pretty much independent of each other rather than building on a common technical contribution - in particular the system building experiment in section 5 does not in any way profit from the corpus analysis or error analysis sections.
>
> Actually, the findings of the analyses in the first two parts are the foundation of our proposed model. For example, we found that the closest antecedents of overt pronouns are always located in subject positions. This pattern is common in nearly all languages showing in Figure 2 (b). Our model benefits from those patterns by incorporating features of UD categories and mention types. We will make a more explicit connection between our analyses and the modeling process in the final version.
>
> > the quantitative analysis stays at the level of what could be done in a few days by taking at face value the corpus annotations rather than looking more deeply into possible confounders.
>
> To be honest, we spent 2 months on the first part of the analysis. We conducted numerous preliminary analyses to understand the datasets first and refined our analysis method in iterative rounds. From these, we present only the most valuable analyses in the paper. Beyond the workload, we would emphasize the analysis methods we employed, such as anaphor-antecedent relation and competing antecedents of pronominal anaphors, which require linguistic expertise and a lot of experience in the field of coreference studies. Please refer to the first answer for the concern of confounders.
>
> > Similarly the error analysis in section 4 states relatively obvious facts that are going to affect any SotA coreference system
>
> Actually, we uncovered new findings through our error analysis. For example, when examining anaphor-antecedent relationships, we found the most frequent UD relation associated with the antecedents of nominal nouns are nominal dependents (e.g., nominal modifier and appositional modifier).  To the best of our knowledge, this paper is the first work to conduct an error analysis from a linguistic perspective across 10 languages. Our approach is supported by substantial experiments and detailed results.
>
> > it presents the rule-based approach of Lee et al 2011 as "Template-based" which I don't find very helpful as a term
>
> We agree with this point and will replace ‘Template-based’ with ‘Rule-based’ in our final version.
>
> > in the subsequent text fails to cleanly separate multilingual (training data in multiple languages) from cross-lingual (training data in one language and testing in another language) tasks or even mixes training methodology (latent structure, joint training) and features (methods with linguistic information).
>
> In the final version, we will reorganize the section of MCR Systems in Related Work by providing categories of training methodologies and approaches that incorporate linguistic information. Under the training methodology category, we will have two additional subcategories: cross-lingual approaches and multilingual approaches.
>
> > Why didn't you include your own system in the error analysis
>
> We will include this error analysis following the method outlined in section 4 in the final version where we will have one more page available.

---

### Meta-Review · Area_Chair_VBMa · 2023-09-10

**Recommendation:** 3

**Metareview:**

The paper provides an in-depth analysis of the CorefUD multi-lingual coreference corpora used in the CRAC2022 shared task, in different linguistic levels (mention, entity and document). Then they did the error analysis on the best-performing system of the shared task to find out the challenges faced by the system. Last they proposed using UD-related features (e.g. UPOS, UD relation) to mitigate the errors. The evaluation on 10% of training data suggests using such features in the baseline system can achieve a marginal improvement of 0.9%. The analysis on the CorefUD corpus and error analysis provides useful insight into multi-lingual coreference resolution. The evaluation, however, is less convincing given that only one system is evaluated on a non-standard test set, which makes it hard to compare with other systems in the future. It is also unclear the reason behind the decision to use CoNLL 2012 F1 instead of the same evaluation metric (CorefUD scorer) used in the CRAC 2022 shared task, This again makes the evaluation non-comparable with the shared task systems even use exactly the same portion of the training set for the test.  The switch between CorefUD 1.1 and CorefUD 1.0 is also a bit confusing, it would be more helpful if the paper stuck to one dataset or at least explained their decision more clearly. There are a few other points that need to be clearly mentioned as well to improve the reproducibility and avoid confusion, e.g. according to the author's response in the discussion the error analysis is done on the dev set. On the UD resources side, (UPOS, UD relations etc.), the author claims during the discussion that those resources that come with the CorefUD are all predicted, but according to the shared task overview paper ( Žabokrtský et al. 2022) in Section 2.1 it mentioned many corpora has UD columns manually annotated (e.g. Prague Dependency Treebank (Czech)) and at the end of Section 2 the organiser mentioned "With some exceptions, if the original resources contained manual annotation of morpho-syntax, it has been kept also in CorefUD. " so it does contain manually annotated UD columns which I would request the author to investigate which corpora they used are with manual UD annotations and make it clear that the results they achieved are not benefit from the gold annotation, e.g. there is no distinguish improvements gained from the datasets with gold annotations.

---

### Decision · Program_Chairs · 2023-10-07

**Decision:**

Accept-Findings

**Comment:**

The paper provides an in-depth analysis of the CorefUD multi-lingual coreference corpora used in the CRAC2022 shared task, in different linguistic levels (mention, entity and document). Then they did the error analysis on the best-performing system of the shared task to find out the challenges faced by the system. Last they proposed using UD-related features (e.g. UPOS, UD relation) to mitigate the errors. The evaluation on 10% of training data suggests using such features in the baseline system can achieve a marginal improvement of 0.9%. The analysis on the CorefUD corpus and error analysis provides useful insight into multi-lingual coreference resolution. The evaluation, however, is less convincing given that only one system is evaluated on a non-standard test set, which makes it hard to compare with other systems in the future. It is also unclear the reason behind the decision to use CoNLL 2012 F1 instead of the same evaluation metric (CorefUD scorer) used in the CRAC 2022 shared task, This again makes the evaluation non-comparable with the shared task systems even use exactly the same portion of the training set for the test.  The switch between CorefUD 1.1 and CorefUD 1.0 is also a bit confusing, it would be more helpful if the paper stuck to one dataset or at least explained their decision more clearly. There are a few other points that need to be clearly mentioned as well to improve the reproducibility and avoid confusion, e.g. according to the author's response in the discussion the error analysis is done on the dev set. On the UD resources side, (UPOS, UD relations etc.), the author claims during the discussion that those resources that come with the CorefUD are all predicted, but according to the shared task overview paper ( Žabokrtský et al. 2022) in Section 2.1 it mentioned many corpora has UD columns manually annotated (e.g. Prague Dependency Treebank (Czech)) and at the end of Section 2 the organiser mentioned "With some exceptions, if the original resources contained manual annotation of morpho-syntax, it has been kept also in CorefUD. " so it does contain manually annotated UD columns which I would request the author to investigate which corpora they used are with manual UD annotations and make it clear that the results they achieved are not benefit from the gold annotation, e.g. there is no distinguish improvements gained from the datasets with gold annotations.